# Untargeted saliva metabolomics by liquid chromatography—Mass spectrometry reveals markers of COVID-19 severity

**Cecile F. Frampas**[1⊙], **Katie Longman**[1⊙], **Matt Spick**[1], **Holly-May Lewis**[1], **Catia D. S. Costa**[2], **Alex Stewart**[3], **Deborah Dunn-Walters**[3], **Danni Greener**[4], **George Evetts**[4], **Debra J. Skene**[3], **Drupad Trivedi**[5], **Andy Pitt**[5], **Katherine Hollywood**[5], **Perdita Barran**[5], **Melanie J. Bailey**[1,2]*

1 Faculty of Engineering and Physical Sciences, University of Surrey, Guildford, United Kingdom, 2 Surrey Ion Beam Centre, University of Surrey, Guildford, United Kingdom, 3 Faculty of Health and Medical Sciences, University of Surrey, Guildford, United Kingdom, 4 Frimley Park Hospital, Frimley Health NHS Trust, Camberley, United Kingdom, 5 Manchester Institute of Biotechnology, University of Manchester, Manchester, United Kingdom

⊙ These authors contributed equally to this work.
* m.bailey@surrey.ac.uk

**Data Availability Statement:** The aligned and annotated LC-MS data matrices used in this work are available at (https://doi.org/10.5281/zenodo.

## Abstract

### Background

The COVID-19 pandemic is likely to represent an ongoing global health issue given the potential for new variants, vaccine escape and the low likelihood of eliminating all reservoirs of the disease. Whilst diagnostic testing has progressed at a fast pace, the metabolic drivers of outcomes–and whether markers can be found in different biofluids–are not well understood. Recent research has shown that serum metabolomics has potential for prognosis of disease progression. In a hospital setting, collection of saliva samples is more convenient for both staff and patients, and therefore offers an alternative sampling matrix to serum.

### Methods

Saliva samples were collected from hospitalised patients with clinical suspicion of COVID-19, alongside clinical metadata. COVID-19 diagnosis was confirmed using RT-PCR testing, and COVID-19 severity was classified using clinical descriptors (respiratory rate, peripheral oxygen saturation score and C-reactive protein levels). Metabolites were extracted and analysed using high resolution liquid chromatography-mass spectrometry, and the resulting peak area matrix was analysed using multivariate techniques.

### Results

Positive percent agreement of 1.00 between a partial least squares–discriminant analysis metabolomics model employing a panel of 6 features (5 of which were amino acids, one that could be identified by formula only) and the clinical diagnosis of COVID-19 severity was achieved. The negative percent agreement with the clinical severity diagnosis was also

6924738). The analytical protocols used are openly available for all researchers to access. The website URL for the protocols is (https://covid19-msc.org/).

**Funding:** The authors would like to acknowledge funding from the Electronics and Physical Sciences Research Council (EPSRC) Impact Acceleration Account for sample collection and processing, as well as EPSRC Fellowship Funding EP/R031118/1, the University of Surrey and Biotechnology and Biological Sciences Research Council (BBSRC) BB/T002212/1. Mass Spectrometry was funded under EPSRC grant EP/P001440/1. https://epsrc.ukri.org/ https://www.ukri.org/councils/bbsrc/ The funders had no role in study design, data collection and analysis, decision to publish, or preparation of the manuscript.

**Competing interests:** The authors have declared that no competing interests exist.

**Abbreviations:** COVID-19, Coronavirus disease 19; CRP, C-reactive protein; HTN, Hypertension; IHD, Ischemic heart disease; KEGG, Kyoto Encyclopedia of Genes and Genomes; LC, Liquid chromatography; LC-MS, Liquid chromatography mass spectrometry; LOOCV, Leave-one-out cross validation; MS, Mass spectrometry; MS/MS or MS2, Tandem mass spectrometry; NPA, Negative percent agreement; PCA, Principal components analysis; PCR, Polymerase chain reaction; PLS-DA, Partial least squares-discriminant analysis; PPA, Positive percent agreement; QC, Quality control; RT-PCR, Reverse transcription polymerase chain reaction; SARS-CoV-2, Severe acute respiratory syndrome coronavirus 2; T2DM, Type 2 diabetes mellitus; VIP, Variable importance in projection.

1.00, leading to an area under receiver operating characteristics curve of 1.00 for the panel of features identified.

## Conclusions

In this exploratory work, we found that saliva metabolomics and in particular amino acids can be capable of separating high severity COVID-19 patients from low severity COVID-19 patients. This expands the atlas of COVID-19 metabolic dysregulation and could in future offer the basis of a quick and non-invasive means of sampling patients, intended to supplement existing clinical tests, with the goal of offering timely treatment to patients with potentially poor outcomes.

## 1. Introduction

The SARS-CoV-2 pandemic has caused a sustained threat to global health since the discovery of the virus in 2019 [1]. Whilst great strides have been made in both treatment and vaccination development [2, 3], the disease has inflicted multiple waves of infection throughout the world during 2020 and into 2021 [4, 5]. COVID-19 has higher fatality rates than seasonal influenza [6], and in addition, new variants are constantly evolving with the potential for either reduced vaccine effectiveness or altered lethality [7]. As a consequence, there is a continuing need for both better understanding of the impact of COVID-19 on the host metabolism as well as for prognostic tests that can be used to triage the high volumes of patients arriving in hospital settings.

Nasopharyngeal swabs followed by polymerase chain reaction (PCR) have been adopted worldwide for SARS-CoV-2 detection. However, supply chains for swabs rapidly collapsed amongst exponential increases in demand for testing, highlighting the urgency for alternative sample types and testing approaches. Furthermore, whilst PCR tests are easily deployable and highly selective for the virus, these approaches yield no prognostic information and cannot easily deliver rapid turnaround at the point of care, for example during a hospital admissions process. In contrast, analyses based on mass spectrometry can be provided in minutes, and have shown promise in the diagnosis of COVID-19 [8]. Furthermore, mass spectrometry instrumentation is often available to hospitals through third party providers or in-house laboratories. Prognostic tests, whilst challenging due to the varied phenotypes that may present themselves [9], could be used to manage demand for hospitalisation and treatment, especially if vaccine escape leads to future waves of severe COVID-19 infection.

Metabolic biomarkers in blood have been identified that carry prognostic information [10–12], but sampling blood is invasive. Our experience in collecting and analysing patient samples is that saliva samples are significantly easier to collect and handle than blood. Blood collection requires trained phlebotomists, causes discomfort to patients and must be spun soon after collection to preserve the metabolome [13]. In contrast, a saliva sample can be donated quickly and painlessly by a patient [14]. Saliva is itself a carrier of the coronavirus [15], and additionally can convey information on wellness via its own characteristic metabolites [16]. To date, saliva is relatively under researched as a biofluid for metabolism analysis. It has been used for breast, pancreatic and also oral cancers [17, 18], and saliva multi-omics has been used to distinguish between COVID-19 inpatients and outpatients [19]. Here we undertook a preliminary and explorative study to investigate the suitability of saliva metabolomics for identifying

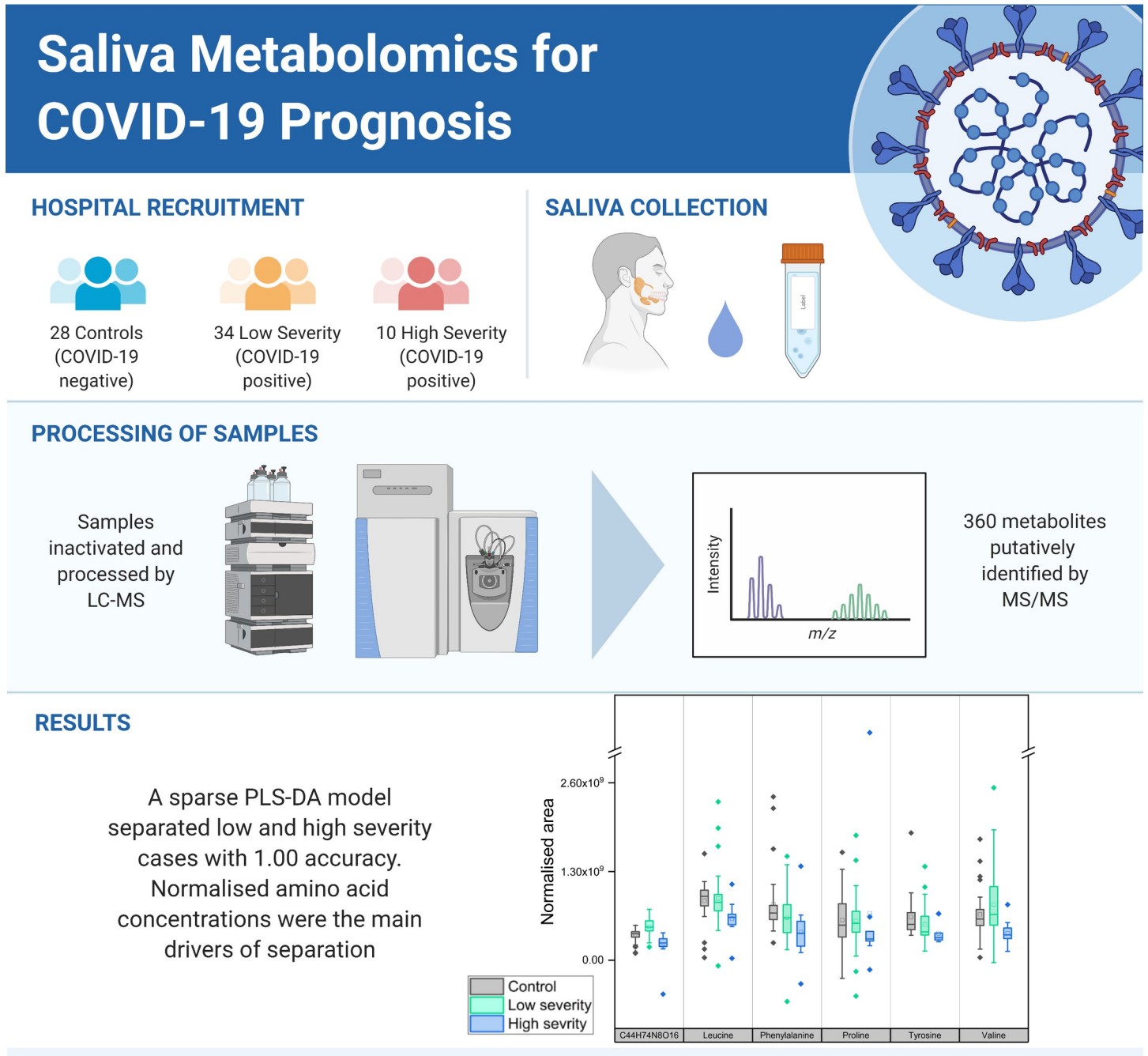

**Fig 1. Workflow summary—Recruitment, processing and results, produced with Biorender.com.**

biomarkers of COVID-19 positivity as well as biomarkers specific to COVID-19 severity within a hospital inpatient cohort (Fig 1).

This work took place as part of the efforts of the COVID-19 International Mass Spectrometry (MS) Coalition [20]. This consortium aims to provide molecular level information on SARS-CoV-2 in infected humans, in order to better understand, diagnose and treat cases of COVID-19 infection. Data related to this work will be stored and fully accessible on the MS Coalition open repository on publication. The website URL is https://covid19-msc.org/

## 2. Materials and methods

### 2.1. Participant recruitment and ethics

Ethical approval for this project (IRAS project ID 155921) was obtained via the NHS Health Research Authority (REC reference: 14/LO/1221). 88 participants were recruited at NHS Frimley NHS Foundation Trust hospitals by researchers from the University of Surrey. Participants were identified by clinical staff to ensure that they had the capacity to consent to the study and were asked to sign an Informed Consent Form, witnessed by two University of Surrey researchers; written / verbal informed consent was obtained from all participants for inclusion in the study, and those that did not have this capacity or who did not provide written consent were not sampled. Consenting participants were categorised by the hospital as either "query COVID" (meaning there was clinical suspicion of COVID-19 infection) or "COVID positive" (meaning that a positive COVID test result had been recorded during their admission). All participants were provided with a Patient Information Sheet explaining the goals of the study.

Inclusion for participants was determined by reverse transcription polymerase chain reaction (RT-PCR) results; participants with an inconclusive RT-PCR test (clinically positive only and/or inconclusive test result, n = 6) or where the time lag between initial RT-PCR test and sampling exceeded fourteen days were excluded (n = 7). These additional exclusion criteria reduced the participant population from 88 to 75.

### 2.2. Sample collection, extraction and instrumental analysis

Patients were sampled immediately upon recruitment to the study in two waves, one between May and August 2020 and the second between October and November 2020. The range in time between symptom onset and saliva sampling ranged from 1 day to > 1 month, an inevitable consequence of collecting samples in a pandemic situation. Subsequently, the population was filtered prior to statistical analysis to exclude patients whose RT-PCR result was greater than 14 days from saliva collection. Each participant provided a sample of saliva by spitting directly into a falcon tube which was placed on ice immediately after collection. Samples were collected between the hours of 9 a.m. and 1 p.m. and transferred on ice from the hospital to the University of Surrey by courier within 4 hours of collection, to minimise changes to salivary metabolites [21]. Once received at University of Surrey, the samples were stored at minus 80°C until analysis.

Alongside saliva collection, metadata for all participants was also collected covering *inter alia* sex, age, comorbidities (based on whether the participant was receiving treatment), the results and dates of COVID PCR tests, bilateral chest X-Ray changes, smoking status, drug regimen, and whether and when the participant presented with clinical symptoms of COVID-19. This included access to medical records, for which consent was given according to the Informed Consent Form described previously. Participants were also sampled for sebum and serum [22]. Values for lymphocytes, CRP and eosinophils were also taken; values obtained within five days of the saliva sampling were recorded. Each participant was attributed a "severity score" in relation to their fitness observations at the time of hospital admission using the metadata collected. We adapted the "mortality scoring" approach of Knight *et al.* [23] to provide a score for symptom severity. This was derived from the sum of the respiratory rate score (with patients scoring 0 for <20, +1 for 20–29 and +2 for ≥30 breaths per min), peripheral oxygen saturation score (%) (0 for ≥30, +2 for <92) and C reactive protein level score (0 for <50 mg/L, +1 for 50–99 mg/L and +2 for ≥100 mg/L). This score ranged from 0 to 6; patients scoring 0 to 3 were attributed low severity and patients scoring 4 to 6 were attributed high severity.

Sample preparation and processing followed the guidelines set out by the COVID-19 Mass Spectrometry Coalition, which included safe handling procedures [13]. Saliva samples were separated into aliquots: 50 μL of saliva was added to 200 μL of ice-cold isopropanol (this also had the effect of deactivating the virus to allow transfer into a lower biological safety level laboratory). The samples were agitated for one hour, sonicated three times for 30 seconds, with resting on ice for 30 seconds between each sonication. Each sample was then left to stand on ice for 30 minutes then centrifuged for 10 minutes at room temperature at 10 000 g before resting on ice. The supernatant was removed and the precipitated protein pellet reserved for future analysis. The supernatant then underwent centrifugal filtration (0.22 μm cellulose acetate) for five minutes at 10,000 g, and the filtered supernatant was then dried under nitrogen and stored at minus 80˚C.

Samples were reconstituted on the day of analysis in 100 μL water:methanol (95:5) with 0.1% formic acid by volume. 10 μL of each sample was set aside for combination in a pooled QC. The samples were analysed over a period of eleven days. Each day consisted of a run incorporating blank injections (n = 2), field blank injections (n = 3), pooled QC injections (n = 6, 3 at the start and finish), as well as QCs to measure instrumental variation and extraction variation (n = 7 and 3 respectively), and 10 participant samples, randomised for positive/negative (n = 3 for each).

## 2.3. Materials and chemicals

The materials and solvents utilised in this study were as follows: 2 mL microcentrifuge tubes (Eppendorf, UK), 0.22 μm cellulose acetate sterile Spin-X centrifuge tube filters (Corning incorporated, USA), 200 μL micropipette tips (Starlab, UK) and Qsert™ clear glass insert LC vials (Supelco, UK). LC-MS grade 2-propanol was used as an inactivation solvent. Optima™ LC-MS grade methanol and water were used as reconstitution solvents and mobile phases. Formic acid was added to the mobile phase solvents at 0.1% (v/v). Solvents were purchased from Fisher Scientific, UK.

## 2.4. Instrumentation and operating conditions

Analysis of samples was carried out using a UltiMate 3000 UHPLC equipped with a binary solvent manager, column compartment and autosampler, coupled to a Q Exactive™ Plus Hybrid Quadrupole-Orbitrap™ mass spectrometer (Thermo Fisher Scientific, UK) at the University of Surrey's Ion Beam Centre. Chromatographic separation was performed on a Waters ACQUITY UPLC BEH C18 column (1.7 μm, 2.1 mm x 100 mm) operated at 55˚C with a flow rate of 0.3 ml min$^{-1}$.

Mobile phase A was water: methanol (v/v 95:5) with 0.1% formic acid, whilst mobile phase B was methanol:water (v/v, 95:5) with 0.1% formic acid (v/v). An injection volume of 5 μL was used. The initial solvent mixture was 2% B for one minute, increasing to 98% B over 16 minutes and held at this level for four minutes. The gradient was finally reduced back to 2% B and held for two minutes to allow for column equilibration. Analysis on the Q-Exactive Plus mass spectrometer was performed with a scan range of $m/z$ 100 to 1 000, and 70,000 mass resolution. MS/MS validation of features was carried out on Pooled QC samples using data dependent acquisition mode and normalised collision energies of 30 and 35 (arbitrary units). Operating conditions are summarised in S1 Table.

## 2.5. Data processing

LC-MS outputs (.raw files) were pre-processed for alignment and peak identification using Compound Discoverer version 3.1 and Freestyle 1.6 (Thermo Fisher Scientific, UK). Peak

picking was set to a mass tolerance ±5 ppm, and alignment to a retention time window of 120 seconds. Missing values were imputed using K-nearest neighbour imputation [24]. Features identified by mass spectrometry were initially annotated using accurate mass match with reference to external databases (explored in parallel; KEGG, Human Metabolome Database, DrugBank, LipidMaps and BioCyc), and then validation was performed using data dependent MS/MS analysis. This process yielded an initial peak:area matrix with 10,700 discrete features. Two criteria were used for inclusion in the final analysis: only those features with identities validated by MS/MS were used, reducing the number of features to 1,874, and 1,514 features that were present in less than 30% of participant samples were excluded. This left 360 features that were used in the analysis. Normalisation was performed using EigenMS in NOREVA for each dataset analysed [25, 26].

## 2.6. Statistical analysis

PCA analyses were conducted in SIMCA (Sartorius Stedim Biotech, France) with additional machine learning conducted in R Studio Version 1.3.959 and MetaboAnalyst [27, 28]. Initial biomarker investigation was carried out using PLS-DA using 5 components and pareto-scaling, maximising separation by mahalanobis distance. Panels of the discriminatory biomarkers were identified by varying the number of features employed but otherwise using the same hyperparameters as for the PLS-DA analyses. Reduced panels were employed to improve robustness, given that when the number of features employed exceeds the number of samples, machine learning can overfit models that lack predictive power [29]. Furthermore, panels emphasising named compounds such as amino acids makes future targeted analysis more straightforward using already-existing assays. Leave-one-out cross-validation was used for model validation test accuracy, sensitivity and specificity; variable importance in projection (VIP) scores were used to assess feature significance alongside p-values and effect sizes (fold count). Batch effects were assessed by PCA analysis of both collection batches (waves one and two) and also instrument and extraction batching by day (in S1 and S2 Figs), showing no clustering by batches.

In prognostic analysis, given the lack of a "gold standard" reference test for whether COVID-19 is likely to be high severity or low severity (as this depends on clinical judgement), positive percent agreement (PPA) between the generated model and a high severity clinical diagnosis was used in preference to sensitivity, which measures the detection of positive instances of a disease relative to a ground truth value. Similarly, negative percent agreement (NPA) between the model and a high severity clinical diagnosis was used in preference to specificity, which measures the detection of positive instances of a disease relative to a ground truth value. In diagnostic analysis, given that RT-PCR tests were available to establish a ground truth, sensitivity and specificity values were calculated alongside diagnostic accuracy.

## 3. Results

### 3.1. Population metadata overview

The study population analysed in this work included 75 participants, comprising 47 participants presenting with a positive COVID-19 RT-PCR test and 28 participants presenting without. Of the positive participants, 10 were classed as presenting with high severity COVID-19, 34 were classed as presenting with low severity COVID-19, and 3 lacked sufficient clinical information for severity scoring. A summary of the metadata is shown in Table 1.

In this study all participants were recruited in a hospital setting with at least potential suspicion of COVID-19 infection; controls were age matched and had similar profiles in terms of gender, oxygen requirements and survival rates. The COVID-19 positive cohort did, however,

present with statistically significant increases in bilateral chest X-ray changes (p-value 0.0009) and levels of eosinophils (p-value 0.002), in agreement with literature observations [23], but not for C-reactive protein (CRP, p-value 0.80). Type 2 diabetes mellitus (T2DM) was more prevalent in the COVID-19 negative population than the positive population, being observed in 36% of COVID-19 negatives versus 30% of high severity COVID-19 patients and 15% of low severity COVID-19 patients, and similar trends of greater comorbidity being seen in the negative population was also true for ischemic heart disease (IHD) and hypertension (HTN). The greater preponderance of underlying comorbidities within the negative population represents a confounding factor.

Within the COVID-19 positive cohort, comorbidities were again age matched, but the high severity grouping had more males (80% male for high severity versus 47% for low severity) and had a statistically significant difference in proportion presenting with hypertension (p-value 0.04) and a statistically significant decrease in eosinophil levels (p-value 0.02). Interestingly, CRP was increased by a 1.5x fold count in high severity participants versus low, but CRP for low severity participants was lower than for COVID-19 negative participants. This can be explained by the fact that CRP is associated with a larger number of comorbidities and patients were only recruited if they had clinical suspicion of COVID-19.

## 3.2. Overview of features identified by liquid chromatography mass spectrometry (LC-MS)

360 features with MS/MS validation were identified as being present in 30% or more of participant samples. Of these 360 features, 36 were identified as related to medical interventions or food and were excluded, leaving 324 for statistical analysis. Of the 324, 38 were annotated by *m/z* value only, 171 were putatively annotated by formula (elemental composition), and 114 were putatively annotated as metabolites, with annotations considered level two as set out by the metabolomics standards initiative (MSI) [30].

## 3.3. Analysis of cohorts by multivariate techniques

Initially separation of COVID-19 positive versus negative participants was tested, as well as separation of COVID-19 high severity and low severity. As shown in Fig 2A, separation for diagnostic purposes was poor by visual inspection and delivered R2Y of 0.78 and Q2Y of 0.18. Leave-one-out cross-validation (LOOCV) provided sensitivity of 0.74 (95% confidence interval of 0.60–0.86) and specificity of 0.75 (0.55–0.89). The most significantly dysregulated identified metabolites (measured by p-value) between positive and negative COVID-19 status are listed in S2 Table.

Fig 2B shows separation for COVID-19 high severity participants versus low severity participants. The optimal separation was found using 5 components. Using leave-one-out cross validation, PPA for COVID-19 high severity was 1.00 (95% confidence interval of 0.69–1.00) and NPA was 1.00 (0.90–1.00), for overall percent agreement with the clinical diagnosis of 1.00 (0.92–1.00).

A volcano plot is shown in Fig 3. Amino acids are highlighted because this class of metabolites was identified as differentiated between high and low severity (see also S3 Table).

In order to improve robustness and reduce overfitting, sparse PLS-DA models were also constructed for the purposes of establishing a smaller panel of metabolites or features capable of discriminating between high and low severity COVID-19 participants. A putative panel comprising Valine, Leucine, Phenylalanine, Tyrosine, Proline and a feature identified putatively only by formula as $C_{44}H_{74}N_8O_{16}$, was capable of discriminating between the two populations with 100% accuracy and AUC of 1.00. This panel of predictive metabolites are

**Table 1. Summary of clinical characteristics by participant cohort.**

| Parameters | Covid-19 Low Severity | Covid-19 High Severity | p-value High vs Low Severity | Covid-19 Negative | p-value Pos vs Neg |
|---|---|---|---|---|---|
| N | 34 | 10 | | 28 | |
| Age (mean, standard deviation; years) | 60 ± 18 | 63 ± 13 | 0.61 | 62 ± 22 | 0.74 |
| Male / Female (n) | 16 / 18 | 8 / 2 | 0.083 | 16 / 12 | 0.26 |
| Treated for Hypertension (n) | 6 | 6 | .041 | 12 | 0.21 |
| Treated for High Cholesterol (n) | 2 | 0 | 1.00 | 6 | .05 |
| Treated for Type 2 Diabetes Mellitus (n) | 5 | 3 | 0.39 | 10 | 0.29 |
| Treated for Ischemic Heart Disease (n) | 1 | 2 | 0.149 | 7 | 0.09 |
| Current Smoker (n) | 1 | 0 | 1.00 | 0 | NA |
| Ex-Smoker (n) | 12 | 5 | 0.71 | 8 | 0.46 |
| Medical Acute Dependency admission (n) | 10 | 6 | 0.26 | 4 | 0.06 |
| Intensive Care Unit admission (n) | 0 | 0 | N/A | 0 | NA |
| Survived Admission (n) | 34 | 8 | 0.048 | 27 | 1.00 |
| Lymphocytes (mean, standard deviation; cells / μL) | 0.8 ± 0.5 | 0.9 ± 0.7 | 0.77 | 1.0 ± 0.5 | 0.302 |
| C-Reactive Protein (mean, standard deviation; mg / L) | 115. ± 85 | 170. ± 83. | 0.075 | 127 ± 105 | 0.80 |
| Eosinophils (mean, standard deviation; 100 / μL) | 0.1 ± 0.1 | 0.0 ± 0.0 | 0.018 | 0.3 ± 0.4 | 0.002 |
| Bilateral Chest X-Ray changes (n) | 15 | 8 | 0.26 | 3 | 0.0009 |
| Continuous Positive Airway Pressure (n) | 1 | 1 | 0.442 | 3 | 0.36 |
| $O_2$ required (n) | 9 | 4 | 0.69 | 8 | 1.00 |

additionally shown as boxplots in Fig 4 below and a complete list of metabolites showing statistically significant differences between high and low COVID-19 severity populations is shown in S3 Table.

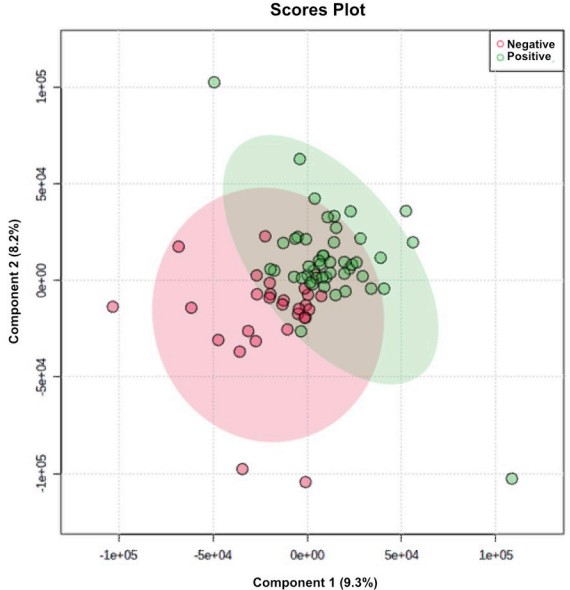
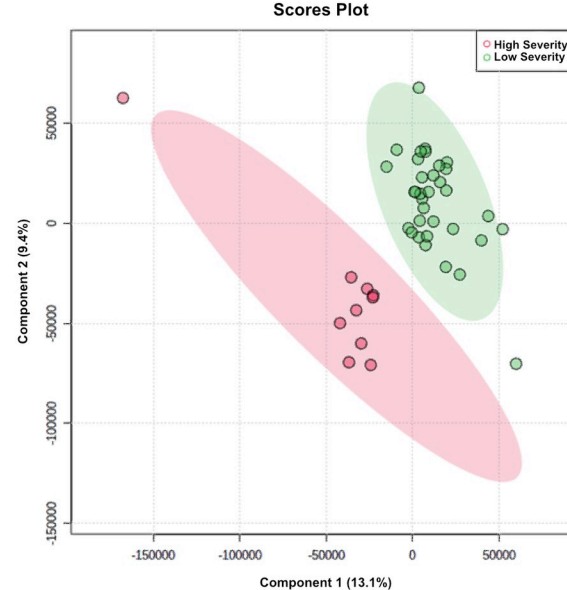

**Fig 2.** Saliva metabolomics analysis for COVID-19 diagnosis and prognosis via LC-MS in positive mode, showing: **A** PLS-DA plot for 75 participants and 324 features, COVID-19 positive / negative. **B** PLS-DA plot for 44 participants and 324 features, high severity / low severity. **C** LOOCV confusion matrix, COVID-19 positive / negative. **D** LOOCV confusion matrix, high severity / low severity.

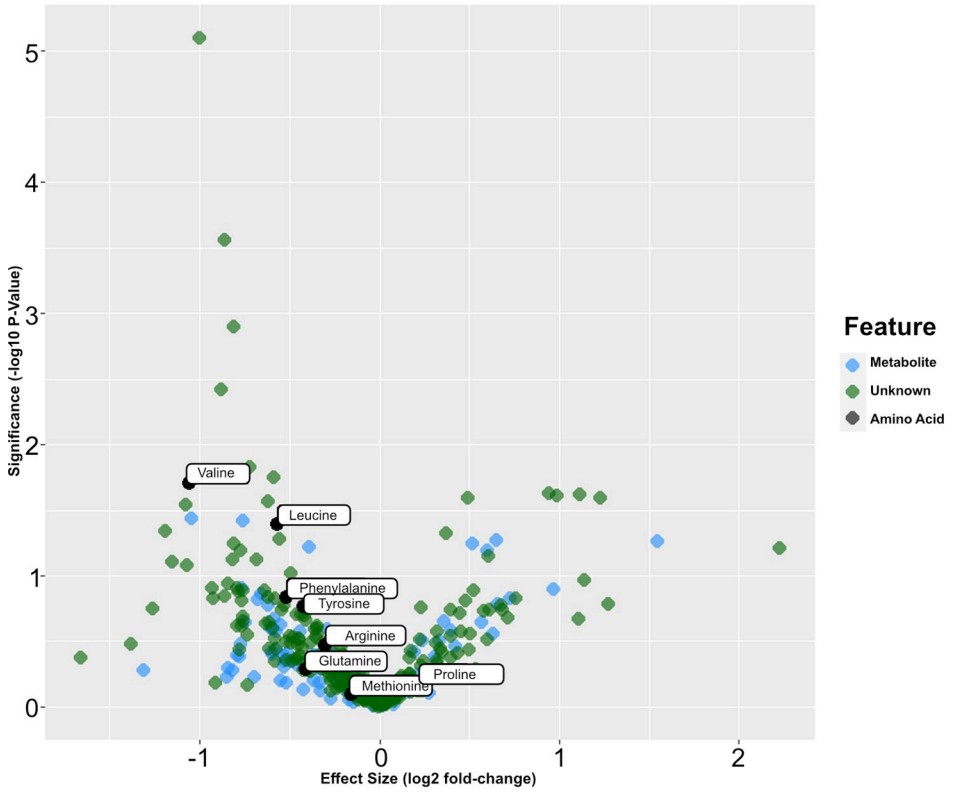

**Fig 3. Volcano plot of statistical significance versus effect size for MS/MS validated features identified in the patient samples.**

### 3.4. Validation set

Since no fully independent prognostic validation set was available, we projected the reduced-feature PLS-DA model obtained for high severity versus low severity participants on to COVID-19 negative participants. Given that these participants should not show features associated with high severity COVID-19, this was considered to offer additional information. The confusion matrix for the results of the projection is shown in Table 2 below.

## 4. Discussion

Whilst age and recruitment venue were well matched (all participants were recruited in a hospital setting including controls), several variables within the metadata illustrate the natural difficulties in experimental design experienced during a pandemic. Age ranges of participants were large, a wide range of comorbidities were present, and the time between symptom onset and saliva sampling ranged from 1 day to > 1 month. This variation in time between symptom onset and saliva sampling was addressed through exclusion of participants whose RT-PCR result was greater than 14 days from study sampling. However, participant recruitment of the most severely affected was limited by ethics approval only covering patients who could give informed consent, thereby precluding the participation of patients with the highest COVID severity. Furthermore, given the small *n* in this pilot study, precision was necessarily low and confidence intervals wide.

In this study, saliva samples were provided under conditions that could be practically achieved in a hospital pandemic setting. This meant no scope for abstinence from food and /

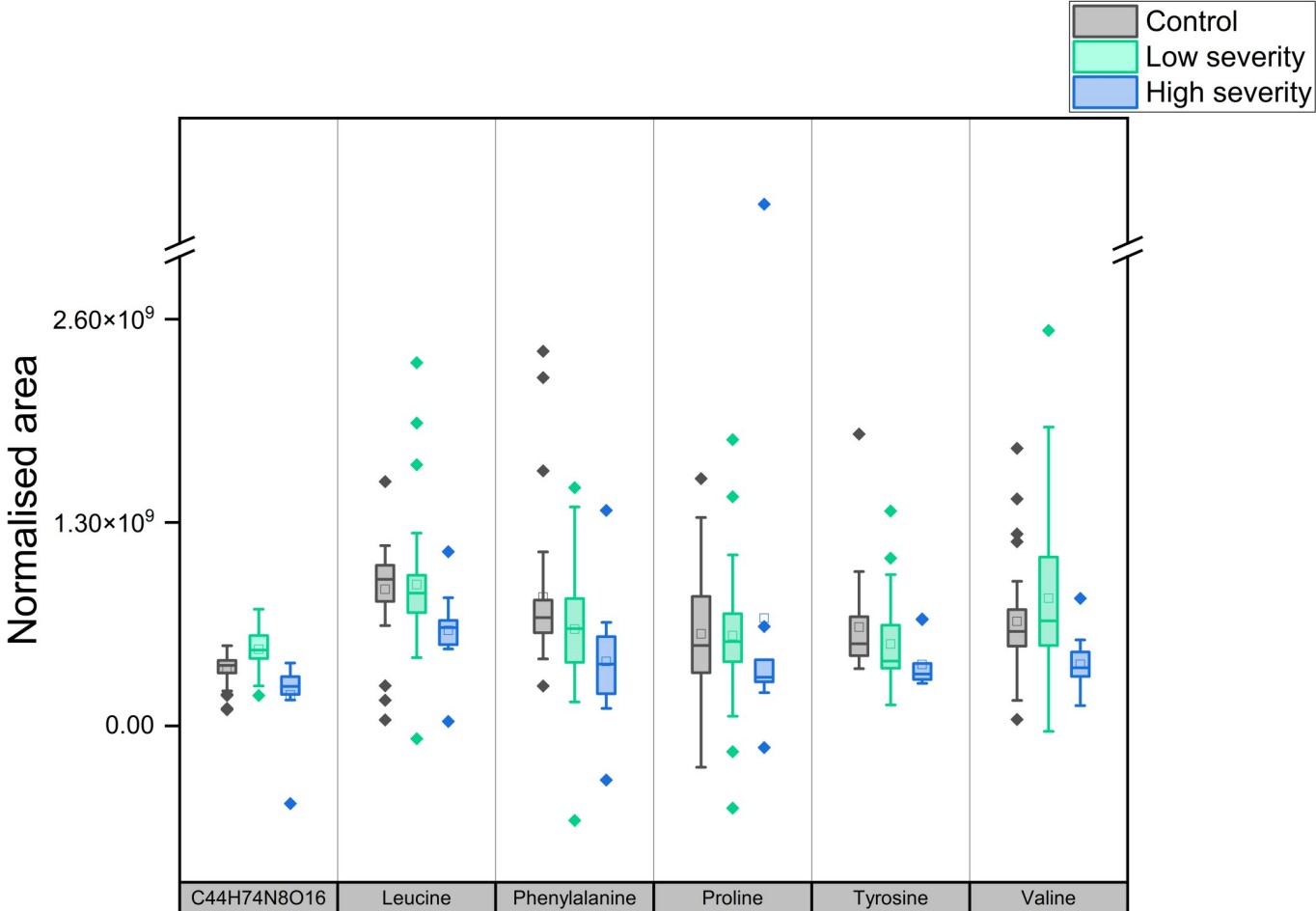

**Fig 4. Boxplots of features selected for ability to differentiate high and low COVID-19 severity (corresponding p-values for high and low severity, left to right: < 0.001, 0.041, 0.051, 0.65, 0.16, and 0.02).**

or drink before saliva sampling, and no prior rinsing of the mouth, leading to potential confounding factors. Diagnostic sensitivity of 0.74 (95% confidence interval of 0.60–0.86) and specificity of 0.75 (0.55–0.89) was considered insufficient to justify further investigation, given that proteomic and serum / plasma based metabolomic diagnostic tests have shown markedly better performance in diagnosing COVID-19 by both meta-analysis and in a recent matched-sample study [8, 31]. Fig 4 illustrates that a more marked separation exists between low severity and high severity, than between hospital-recruited controls and low severity. We hypothesise that mild COVID-19 causes more limited alteration of the salivary metabolome versus controls, especially given that the controls in this work were recruited in a hospital setting with similar symptoms to COVID-19. The data presented here suggest that salivary dysregulation specific to COVID-19 (and not indicative of general poor health) only reaches clearly

**Table 2. Confusion matrix for reduced-feature PLS-DA model projected on to COVID-19 negative participants.**

|  | COVID-19 negative participants |
|---|---|
| **Reduced-feature PLS-DA model result: High Severity** | 1 |
| **Reduced-feature PLS-DA model result: Low Severity** | 27 |

identifiable levels at greater levels of COVID-19 severity, at least using the uncontrolled, but pragmatic sampling approach described in this work.

Superior differentiation by multivariate analysis was, however, achieved in relation to COVID-19 severity. The reduced-feature PLS-DA model showed separation of High Severity COVID-19 positive participants from Low Severity COVID-19 positive participants, with PPA and NPA of 1.00 by LOOCV. Furthermore, whilst not a true independent validation set, projecting the reduced-feature PLS-DA model on to COVID-19 negative participants showed that the model classified 97% of them as "low risk", i.e. that the characteristic levels of markers associated with high severity were not associated with low severity in COVID-19 negative participants.

A number of identified metabolites showed statistically significant differences between the high and low severity participants. As shown in Fig 4, amino acids constituted the class of metabolites seeing the most changes between high and low severity, similar to literature observations of changes in either amino acids or ratios of amino acids in serum or plasma. Encouragingly for clinical application, in this work a reduced feature panel of just six features was still capable of discriminating between high and low severity COVID-19 participants, with five of the six features being amino acids (all downregulated in the saliva of high-severity participants). One previous study found in contrast that salivary myo-inositol and 2-pyrrolidineacetic acid were capable of distinguishing an inpatient cohort from an outpatient cohort [19], rather than amino acids, but all recruitment in this work took place in a hospital setting, i.e. the results shown herein represent separation based on severity within the inpatient cohort.

A number of limitations in this study should be acknowledged. In this work, we were unable to standardise the saliva collection by asking patients to rinse the mouth or abstain from eating, due to health and safety considerations, Additionally, we were unable to access patients immediately after admission to hospital, meaning that sample collection took place 1 day– 1 month after admission. Further, high resolution mass spectrometry was only performed in positive mode, due to competing demands for participant samples. Analysis in both positive mode and negative mode could have identified additional significant features. The analysis was also untargeted, and so lacked the use of internal standards that would be available in a targeted assay. As an untargeted analysis, many features were putatively identified, resulting in a noisy dataset for machine learning. In addition, the supervised multivariate analysis used in this work can lead to overfitting and false discovery, especially given the relatively small numbers of participants recruited in this work. Validation of these results in a larger and more balanced study cohort is required, using a standardised approach for assessing COVID-19 severity, when such an approach is universally accepted. Furthermore, future studies may need to take account of biomarker changes with new variants, as these have been found to be dependent upon collection wave [32].

It should be noted, however, that whilst this work was untargeted for discovery purposes, the features selected for the PLS-DA panel (valine, leucine, phenylalanine, tyrosine, proline and $C_{44}H_{74}N_8O_{16}$) were on average present in 85% of participant samples and targeted methods are available that can reliably quantify amino acids for future investigation [33]. Therefore, whilst this is a preliminary and exploratory study, we see these results as encouraging first evidence for distinctive changes in the salivary metabolome of hospitalised individuals with severe COVID-19. We believe that saliva has potential to add to understanding of the progression and severity of COVID-19, providing evidence that the salivary metabolome is disrupted, and more generally illustrating the potential for saliva as a biofluid for investigating dysregulated metabolism related to infectious diseases.

## Supporting information

**S1 Fig. Clustering of patient samples according to extraction batch.** Principal component analysis of each patient sample (circles) and batch QC's (squares), coloured according to extraction batch, showing no significant clustering of patient samples according to extraction batch.
(DOCX)

**S2 Fig. Principal component analysis of each patient sample and run QC. Figure shows low levels of QC variation according to position in the run sequence.**
(DOCX)

**S3 Fig. Principal component analysis for 75 participants and 324 features, COVID-19 positive / negative, LC-MS analysis in positive mode.**
(DOCX)

**S4 Fig. Principal component analysis for 44 participants and 324 features, high severity / low severity, LC-MS analysis in positive mode.**
(DOCX)

**S1 Table. Operating conditions of the mass spectrometer used in this research.**
(DOCX)

**S2 Table. Distinctive features between COVID-19 positive and negative.**
(DOCX)

**S3 Table. Distinctive features between COVID-19 high severity and low severity.**
(DOCX)

## Acknowledgments

The authors acknowledge Samiksha Ghimire from Groningen Medical School for translation of participant information sheets and consent forms into Nepalese, as well as Kyle Saunders of the University of Surrey for access to batch controls. The authors are additionally grateful to Thanuja Weerasinge (Jay), Manjula Meda, Chris Orchard and Joanne Zamani of Frimley Park NHS Foundation Trust for their help with ethics approvals and access to hospital patients.

## Author Contributions

**Conceptualization:** Melanie J. Bailey.

**Data curation:** Matt Spick, Holly-May Lewis, Catia D. S. Costa.

**Formal analysis:** Cecile F. Frampas, Katie Longman.

**Funding acquisition:** Melanie J. Bailey.

**Investigation:** Cecile F. Frampas, Katie Longman, Danni Greener, George Evetts, Debra J. Skene, Melanie J. Bailey.

**Methodology:** Cecile F. Frampas, Katie Longman, Holly-May Lewis, Catia D. S. Costa, Alex Stewart, Deborah Dunn-Walters, Danni Greener, George Evetts, Debra J. Skene, Drupad Trivedi, Andy Pitt, Katherine Hollywood, Perdita Barran.

**Project administration:** Alex Stewart, Deborah Dunn-Walters, Melanie J. Bailey.

**Resources:** Danni Greener, George Evetts, Debra J. Skene, Melanie J. Bailey.

**Software:** Matt Spick.

**Supervision:** Melanie J. Bailey.

**Visualization:** Matt Spick.

**Writing – original draft:** Cecile F. Frampas, Katie Longman.

**Writing – review & editing:** Matt Spick, Alex Stewart, Deborah Dunn-Walters, Debra J. Skene, Melanie J. Bailey.

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
