## [Decision Letter · Decision Letter 0]

12 Jul 2022

PONE-D-22-09481Untargeted saliva metabolomics by liquid chromatography - mass spectrometry reveals biomarkers of COVID-19 severityPLOS ONE

Dear Dr. Matt P. Spick,

Thank you for submitting your manuscript to PLOS ONE. After careful consideration, we feel that it has merit but does not fully meet PLOS ONE’s publication criteria as it currently stands. Therefore, we invite you to submit a revised version of the manuscript that addresses the points raised during the review process.

We look forward to receiving your revised manuscript.

Kind regards,

Tommaso Lomonaco, Ph.D

Academic Editor

PLOS ONE

Journal Requirements:

2. Please amend either the abstract on the online submission form (via Edit Submission) or the abstract in the manuscript so that they are identical.

4. Please expand the acronym “EPSRC, BBSRC” (as indicated in your financial disclosure) so that it states the name of your funders in full.

Reviewers' comments:

Reviewer's Responses to Questions

**Comments to the Author**

1. Is the manuscript technically sound, and do the data support the conclusions?

Reviewer #1: Yes

Reviewer #2: Yes

2. Has the statistical analysis been performed appropriately and rigorously? 

Reviewer #1: No

Reviewer #2: Yes

3. Have the authors made all data underlying the findings in their manuscript fully available?

Reviewer #1: Yes

Reviewer #2: Yes

4. Is the manuscript presented in an intelligible fashion and written in standard English?

Reviewer #1: Yes

Reviewer #2: Yes

5. Review Comments to the Author

Reviewer #1: In this paper, the untargeted profiling of the salivary metabolome in COVID-19 patients is interestingly presented. This topic seems to be really appealing in this field and the work takes place as part of the “COVID-19 International Mass Spectrometry (MS) Coalition” attempt.

I would like to suggest some comments to improve the manuscript and clarify some points:

Main general comments:

Even considering the natural difficulties experienced in patient recruitment and sample collection during a pandemic, the pilot study described in this work has not been designed in a systematic manner. This is evident from the lack of detailed instructions for saliva sample collection and a variable timeframe occurring between symptom onset and saliva sampling (from 1 day to > 1 month). These conditions represent substantial confounding factors when assessing the predictive role of salivary metabolites in COVID-19. A scattered sample collection generally produces random results and wrong data interpretation. Considering the unsystematic approach and the small population size (10 high severity, 34 low-severity, 28 COVID-19 negative), it is not appropriate to describe your approach as a prognostic tool to assess COVID-19 severity. The proposed pilot study should be described as a preliminary and explorative study to hypothesize the role of salivary metabolome as a predictor of COVID-19 severity, more than a study to identify biomarkers. As mentioned in the discussion section, this pilot study should be confirmed through a wide-scale prospective observational study, mainly focused on a subset of selected chemicals analysed in targeted mode. Considering all these aspects, please restate properly the title and the objective of the work/conclusions in the abstract session. Be in line with the considerations included in the discussion section.

Specific comments:

Abstract:

Line 35: Please add the list of clinical descriptors between brackets.

Lines 36-37: Please consider changing “mass spectrometry” with “high-resolution mass spectrometry”

Please delete the colon between peak and area.

Line 41: Specify the sixth feature

Lines 44-45: Please add the ROC curve to the Supplementary information section.

Lines 48-49: Unfortunately, the criteria to define low/high severity in COVID-19 are not standardized. Guidelines are not unique worldwide and, as you said, the threshold between low and high depends on clinical judgement. In my opinion, it is recommended to wait for a wider population size and more standardised criteria to define COVID-19 severity before proposing predictive models for prognostic purposes.

Introduction:

Line 77: “In contrast, tests based on mass spectrometry can be provided in minutes..” Are you talking about on-line instrumentation? Unfortunately, when dealing with off-line approaches and sample treatment, the time is actually longer. Please, restate or insert a reference.

Line 78: “Furthermore, mass spectrometry instrumentation is often available in hospital pathology laboratories.” Unfortunately, this is not true. MS is not among the facilities commonly found in hospitals/clinics. Please restate or insert a reference.

Line 81: “especially should vaccine escape lead to..”. Unclear. Please, restate.

Line 87: “and must be spun soon after collection to preserve the metabolome”. Please add a reference.

Line 88: “a saliva sample can be donated quickly and painlessly by a patient” Add the following reference on saliva collection (doi.org/10.1016/j.trac.2019.115781).

Line 89: “Saliva is itself a carrier of the coronavirus” This is not a good thing from a biosafety point of view. How did you ensure a safe collection/manipulation of the saliva specimen? Did you follow any standardized protocol? Please, insert some details on this important aspect.

Lines 89-90: “information via its own characteristic metabolites. [13]..” What do you mean? Please, clarify.

Materials and methods:

Lines 123-124: “saliva sampling ranged from 1 day to > 1 month..” This aspect could represent a critical issue especially for the main objective of the study, i.e. identifying prognostic markers. The levels and, thus, the predictivity are influenced by the timeframe occurring between symptoms/collection. I kindly suggest organising the next study by collecting samples 2-3 days maximum after the occurrence of symptoms for all the investigated subjects.

Line 125: “Each participant provided a sample of saliva by spitting directly into a falcon tube which was placed on ice immediately after collection.” What about additional details on saliva collection protocol? Same period of the day? Morning or afternoon? No food, smoke, beverages before sampling? Mouth rinsing? The straightforward application of a defined protocol in the clinical routine is not easy, but an unsystematic sample collection (especially for a biofluid as saliva) leads to random data acquisition.

Lines 141-142: Supplementary oxygen, spontaneous and/ or assisted ventilation? Be specific.

Line 153: Which temperature were centrifuged the samples?

Line 158: Considering the reconstitution with 100 μL water:methanol (95:5), you are diluting your sample 1:2 v/v. Are you loosing this way all the information related to trace/ultra-trace components? Would be better a clean-up and pre-concentration step?

Line 186: Considering the sophisticated instrumentation available in your lab, why not choosing a 2 ppm accuracy?

Line 189: Did you work both in positive and negative mode? If not, why not considering a sample characterization even in negative mode? Why choosing 70 000 as resolution?

Lines 196-197: Did you employ all the cited databases sequentially? Specify. What about METLIN?

Line 208: Since we are referring to an explorative pilot study characterized by a small sample-size for each group (≤30), I’d rather use an unsupervised technique as PCA instead of PLS-DA. The use of PLS would be more indicated when dealing with a restricted panel of marker (selected by this preliminary work, confirmed through pure standard and analysed in targeted mode) and a wider population to build up robust predictive models.

Just out of curiosity, please furnish the PCA plot for both positive/negative and low/high severity conditions.

Results:

Table 1: Please insert the correct digits for the age and C-reactive protein data.

Line 243: “controls were age matched and had similar profiles in terms of gender, oxygen requirements and survival rates”. Do controls present oxygen requirements similar to both low/high severe patients?

Lines 258-260: Do you have an explanation for this unexpected behaviour?

Lines 272-273: Did you characterise your samples both in positive and negative mode? Figure 2A/2B are referred to positive acquisition mode. Isn’t it? How many features? Specify.

Line 280: “The optimal separation was found using 5 components”. By using just the first two PLS component, we can explain 22.5% of the variation in the response variable. This value is quite low, suggesting the presence of lot of noise in your raw data instead of valuable information. Please add a comment.

Line 294: Please change “COVID” with “COVID-19”.

Line 296: Insert a p-value in the text just to understand what you mean by “most disturbed”.

Lines 298-299: “MS/MS validated features separating..” Specify how many features are you referring to.

Line 311-313: “It was decided to project the PLS-DA model obtained for high severity versus low severity participants on to COVID-19 negative participants”. I disagree. The population of the negative participants may not be representative of people infected by SARS-CoV-2 and thus may not be a proper test set. Please justify your decision.

Discussion:

Line 325: Please see comments lines 123-124.

Lines 336-339: How can you explain the presence of a more marked difference between low/high severity than positive/negative subjects (where high severity is not so high as you mentioned, because most of the very severe patients were excluded from the study)?

Please add some references about recent experimental works which already suggested differences between low and high severity in COVID-19 based on biomarkers (as the most famous markers of inflammation) (doi.org/10.1002/rmv.2146;
doi.org/10.1093/nsr/nwaa086;
doi.org/10.1016/j.freeradbiomed.2022.01.021) to strengthen your results.

Lines 340-342. I disagree. The alteration of plasmatic metabolome in infected people is generally dramatically marked when compared to healthy subjects (probably this is reflected even in saliva). Please delete this sentence or confirm it by adding some references.

Line 349: “the features associated with high severity were present neither in low severity nor in COVID-19 negative participants.” Were they absent or still present but characterised by lower levels?

Line 350: How many? Only in positive mode? Please specify.

Line 357: Have you a biochemical explanation for this down-regulation?

Lines 364-365: I fully agree. The best choice would be to post-pone the use of predictive models to future studies characterized by a wider cohort of subjects and a selected panel of chemicals fully identified and quantified in targeted mode.

Reviewer #2: Saliva samples were collected from COVID-19 suspected patients and confirmed by RT-PCR. Then the samples were analyzed for metabolites by Liquid Chromatography Mass Spectroscopy (LC-MS) and correlated to COVID-19 infection status of the patients. The metabolites and patient data (sex, age, comorbidities (whether any treatment , the results and dates of COVID PCR tests, bilateral chest X-Ray changes, smoking status, drug regimen, and whether and when the participant presented with clinical symptoms of COVID-19 etc.). Authors concluded that COVID-19 severity was related to the alteration of the salivary metabolome.

6. PLOS authors have the option to publish the peer review history of their article (what does this mean?). If published, this will include your full peer review and any attached files.

Reviewer #1: No

Reviewer #2: **Yes: **Veli Cengiz Ozalp

---

## [Author Response · Author response to Decision Letter 0]

31 Aug 2022

Please note that these responses have also been uploaded as a file.

Response to reviewers

We are grateful to the reviewers for taking their time to review our manuscript. Please find herein our point-by-point response to each of the reviewers.

Editorial comments

We have resubmitted in accordance with file naming requirements.

2. Please amend either the abstract on the online submission form (via Edit Submission) or the abstract in the manuscript so that they are identical.

We have resubmitted identical Abstracts.

We have amended the manuscript to include a direct link to the Zenodo data repository where the complete dataset is saved for open access by all.

https://doi.org/10.5281/zenodo.6924738

4. Please expand the acronym “EPSRC, BBSRC” (as indicated in your financial disclosure) so that it states the name of your funders in full. This information should be included in your cover letter; we will change the online submission form on your behalf.

This has been expanded in the manuscript (financial disclosure).

We have deleted the section “Ethics Declaration” so that now the only ethics statement is in the methods section

This has been updated

Reviewer 1 comments

In this paper, the untargeted profiling of the salivary metabolome in COVID-19 patients is interestingly presented. This topic seems to be really appealing in this field and the work takes place as part of the “COVID-19 International Mass Spectrometry (MS) Coalition” attempt.

I would like to suggest some comments to improve the manuscript and clarify some points:

Main general comments:

Even considering the natural difficulties experienced in patient recruitment and sample collection during a pandemic, the pilot study described in this work has not been designed in a systematic manner. This is evident from the lack of detailed instructions for saliva sample collection and a variable timeframe occurring between symptom onset and saliva sampling (from 1 day to > 1 month). These conditions represent substantial confounding factors when assessing the predictive role of salivary metabolites in COVID-19. A scattered sample collection generally produces random results and wrong data interpretation. Considering the unsystematic approach and the small population size (10 high severity, 34 low-severity, 28 COVID-19 negative), it is not appropriate to describe your approach as a prognostic tool to assess COVID-19 severity. The proposed pilot study should be described as a preliminary and explorative study to hypothesize the role of salivary metabolome as a predictor of COVID-19 severity, more than a study to identify biomarkers. As mentioned in the discussion section, this pilot study should be confirmed through a wide-scale prospective observational study, mainly focused on a subset of selected chemicals analysed in targeted mode. Considering all these aspects, please restate properly the title and the objective of the work/conclusions in the abstract session. Be in line with the considerations included in the discussion section.

We agree with the reviewer’s identifications of the challenges we faced. We have amended the text to state that this is a preliminary and explorative study, as suggested, and rather than discussing saliva metabolomics in terms of its ability to act as a prognostic tool, have rephrased the text to refer to saliva metabolomics in terms of its ability to differentiate between severe and non-severe cases of COVID-19. These points are also reflected in the changes made in response to the detailed comments below.

Specific comments:

Abstract:

Line 35: Please add the list of clinical descriptors between brackets.

This has been amended

Lines 36-37: Please consider changing “mass spectrometry” with “high-resolution mass spectrometry”

We have made this change.

Please delete the colon between peak and area.

We have made this change.

Line 41: Specify the sixth feature

This has been amended

Lines 44-45: Please add the ROC curve to the Supplementary information section.

Given that the area under receiver operating curve is 1.00 we think that adding a ROC curve will not add value to the reader.

Lines 48-49: Unfortunately, the criteria to define low/high severity in COVID-19 are not standardized. Guidelines are not unique worldwide and, as you said, the threshold between low and high depends on clinical judgement. In my opinion, it is recommended to wait for a wider population size and more standardised criteria to define COVID-19 severity before proposing predictive models for prognostic purposes.

We agree with the reviewer that there is no accepted method for classification of COVID-19 severity, and this is a current research topic. However, there are several published precedents of using blood for providing a prognostic information for COVID-19. We have amended the discussion to explain this limitation.

Introduction:

Line 77: “In contrast, tests based on mass spectrometry can be provided in minutes..” Are you talking about on-line instrumentation? Unfortunately, when dealing with off-line approaches and sample treatment, the time is actually longer. Please, restate or insert a reference.

We appreciate the reviewer’s concern here, this refers to off-line instrumentation (both for PCR and mass spectrometry). We appreciate that the run time on the instrument is not the same as the time to send the sample to a laboratory and return a test result. However this is the same in both cases, so it is correct that mass spectrometry is faster than RT-PCR. There is a fundamental difference between mass spectrometry and RT-PCR run times, because the amplification step in RT-PCR takes several hours due to the temperature cycling. The reference cited in the text gives more detail on both approaches. We have adapted to the test to specify that the analysis (rather than the test) is provided in minutes.

Line 78: “Furthermore, mass spectrometry instrumentation is often available in hospital pathology laboratories.” Unfortunately, this is not true. MS is not among the facilities commonly found in hospitals/clinics. Please restate or insert a reference.

Our experience in the UK is that it is, but we agree with the reviewer that this is not necessarily the case internationally, or for smaller hospitals. We have rephrased to show that a mass spectrometry service is normally at least available (perhaps from a third party).

Line 81: “especially should vaccine escape lead to..”. Unclear. Please, restate.

This has been amended

Line 87: “and must be spun soon after collection to preserve the metabolome”. Please add a reference.

We have added a reference to the MS Coalition protocols for biofluid collection to meet this concern.

Line 88: “a saliva sample can be donated quickly and painlessly by a patient” Add the following reference on saliva collection (doi.org/10.1016/j.trac.2019.115781).

This reference has been added to the manuscript.

Line 89: “Saliva is itself a carrier of the coronavirus” This is not a good thing from a biosafety point of view. How did you ensure a safe collection/manipulation of the saliva specimen? Did you follow any standardized protocol? Please, insert some details on this important aspect.

The saliva handling is carried out under the guidance given by the mass spectrometry coalition, which is referenced in the paper (reference 22). The protocols also explain the health and safety aspects of handling saliva from hospitalised patients. This has been added to the methods section.

Lines 89-90: “information via its own characteristic metabolites. [13]..” What do you mean? Please, clarify.

The text has been expanded to explain

Materials and methods:

Lines 123-124: “saliva sampling ranged from 1 day to > 1 month..” This aspect could represent a critical issue especially for the main objective of the study, i.e. identifying prognostic markers. The levels and, thus, the predictivity are influenced by the timeframe occurring between symptoms/collection. I kindly suggest organising the next study by collecting samples 2-3 days maximum after the occurrence of symptoms for all the investigated subjects.

Patients whose RT-PCR result was greater than 14 days from saliva sampling were removed from the sample set for statical analysis. This has now been clarified within the methods section. However, we do agree with this point, and have made this a recommendation for future studies – see our discussion section.

Line 125: “Each participant provided a sample of saliva by spitting directly into a falcon tube which was placed on ice immediately after collection.” What about additional details on saliva collection protocol? Same period of the day? Morning or afternoon? No food, smoke, beverages before sampling? Mouth rinsing? The straightforward application of a defined protocol in the clinical routine is not easy, but an unsystematic sample collection (especially for a biofluid as saliva) leads to random data acquisition.

Samples were collected at the same time of day (morning), but it was not possible to control rinsing (due to health and safety restrictions at the time), and consumption of beverages, due to ethical considerations – we were not allowed to control the dietary intake of our patients. We agree that a consistent sampling procedure would be ideal, but unfortunately, we had operational constraints. We decided to take the pragmatic approach and work around these. Since these constraints were likely to be faced in any practical implementation of this test, we aimed at finding markers for COVID severity that were robust enough not to be perturbed by the sample collection method. We have made several edits to the manuscript to explain this better.

Lines 141-142: Supplementary oxygen, spontaneous and/ or assisted ventilation? Be specific.

This information was not recorded as it was not required to generate the respiratory rate score. The score was generated using measurements of breaths per minute and blood oxygen saturation on admission to A&E. 

Line 153: Which temperature were centrifuged the samples?

They were centrifuged at room temperature. This has been added to the text.

Line 158: Considering the reconstitution with 100 μL water:methanol (95:5), you are diluting your sample 1:2 v/v. Are you loosing this way all the information related to trace/ultra-trace components? Would be better a clean-up and pre-concentration step?

We took the method published by the International COVID-19 mass spectrometry coalition, which aimed at providing standardised methods that could be adopted by multiple laboratories to enable data sharing. We agree that the method could have been better optimised , but chose to use this method to enable the comparison of our data with that from other laboratories around the world.

Line 186: Considering the sophisticated instrumentation available in your lab, why not choosing a 2 ppm accuracy?

Again, we agree that the method used could have been better optimised. However, for this work, we prioritised using the standardised method which would allow for global comparison of COVID-19 data. 

Line 189: Did you work both in positive and negative mode? If not, why not considering a sample characterization even in negative mode? Why choosing 70 000 as resolution?

We ran a set of pooled saliva samples using the method and found that positive mode identified a greater number of features than negative mode. This is also the case in literature. We do, of course, recognise that utilising both modes will always be additive, but we were constrained on two fronts. First, saliva samples were scarce and we faced competing demands for the patient samples (proteomics, sequencing). Second, the LC-MS method employed was lengthy and took two weeks of instrumental run-time and instrumental availability meant that we were unable to run both positive and negative in the time available. We have adjusted the text to recognise this as a limitation.

Lines 196-197: Did you employ all the cited databases sequentially? Specify. What about METLIN?

These are employed in parallel by the Compound Discoverer software. Metlin was not used but we believe the databases listed are comprehensive.

Line 208: Since we are referring to an explorative pilot study characterized by a small sample-size for each group (≤30), I’d rather use an unsupervised technique as PCA instead of PLS-DA. The use of PLS would be more indicated when dealing with a restricted panel of marker (selected by this preliminary work, confirmed through pure standard and analysed in targeted mode) and a wider population to build up robust predictive models.

We agree with the reviewer that unsupervised approaches are important in the exploratory stage. Supervised analyses are also helpful in identifying variability in the feature set specifically related to the condition being investigated, but with small n and in a pilot study, we agree are prone to overfitting and should not be over-interpreted until validated in a more robust and wider population analysis. We have extended the discussion of limitations to address this specific point.

Just out of curiosity, please furnish the PCA plot for both positive/negative and low/high severity conditions.

This has been added to the supplementary materials

Results:

Table 1: Please insert the correct digits for the age and C-reactive protein data.

This has been changed

Line 243: “controls were age matched and had similar profiles in terms of gender, oxygen requirements and survival rates”. Do controls present oxygen requirements similar to both low/high severe patients?

Oxygen requirements were slightly skewed towards the high severity group at 40% versus 22% in low severity patients. This is as expected in severe cases of COVID-19 where low blood oxygen level is associated with poor patient outcomes. 

Lines 258-260: Do you have an explanation for this unexpected behaviour?

This has been adapted in the text. We believe the reason is that CRP is not specific to COVID and all patients had clinical suspicion of COVID-19 infection, as discussed.

Lines 272-273: Did you characterise your samples both in positive and negative mode? Figure 2A/2B are referred to positive acquisition mode. Isn’t it? How many features? Specify.

The reviewer is correct this is positive mode only, as discussed above. Response to be confirmed - Added

Line 280: “The optimal separation was found using 5 components”. By using just the first two PLS component, we can explain 22.5% of the variation in the response variable. This value is quite low, suggesting the presence of lot of noise in your raw data instead of valuable information. Please add a comment.

We agree and we have added this limitation to the discussion 

Line 294: Please change “COVID” with “COVID-19”.

We have made this change.

Line 296: Insert a p-value in the text just to understand what you mean by “most disturbed”.

The text has been amended

Lines 298-299: “MS/MS validated features separating..” Specify how many features are you referring to.

We have edited the text of this caption – this is all the MS/MS validated features.

Line 311-313: “It was decided to project the PLS-DA model obtained for high severity versus low severity participants on to COVID-19 negative participants”. I disagree. The population of the negative participants may not be representative of people infected by SARS-CoV-2 and thus may not be a proper test set. Please justify your decision.

This was done in order to test the model. We agree with the reviewer that there is a possibility that the negative participants and low severity participants are not representative and have acknowledged in the text that this does not constitute the ideal validation strategy. However, as stated at the beginning of the results section, the two populations are age and gender matched, and in the same hospital environment. In addition, we found very little separation between COVID-19 positive and negative patients. Therefore we have no evidence to suggest that these populations are not representative.

Discussion:

Line 325: Please see comments lines 123-124.

The text has been amended to specify removal of patients whose PCR was greater than 14 days from saliva sampling. 

Lines 336-339: How can you explain the presence of a more marked difference between low/high severity than positive/negative subjects (where high severity is not so high as you mentioned, because most of the very severe patients were excluded from the study)?

We hypothesise that at low severity, the differences versus our controls are modest – we also note that our controls are also hospitalised inpatients with clinical symptoms suggestive of COVID-19 infection. This may reduce the differentiation between mild cases and equivalently sick patients regarding e.g. biomarkers relating to generalised inflammation. We have expanded the discussion of this point in the Discussion.

Please add some references about recent experimental works which already suggested differences between low and high severity in COVID-19 based on biomarkers (as the most famous markers of inflammation)

(doi.org/10.1002/rmv.2146;
doi.org/10.1093/nsr/nwaa086;
doi.org/10.1016/j.freeradbiomed.2022.01.021)

to strengthen your results.

We agree that these references are helpful and have added them to the manuscript.

Lines 340-342. I disagree. The alteration of plasmatic metabolome in infected people is generally dramatically marked when compared to healthy subjects (probably this is reflected even in saliva). Please delete this sentence or confirm it by adding some references.

We agree and have modified the text to improve clarity – our original comment was intended to apply only to the salivary metabolome. In homeostatically regulated biofluids infection does have a more marked influence.

Line 349: “the features associated with high severity were present neither in low severity nor in COVID-19 negative participants.” Were they absent or still present but characterised by lower levels?

The markers were not absent in the negative patients – their pattern was more like the low severity patients, and the text has been modified to explain this.

Line 350: How many? Only in positive mode? Please specify.

Only in positive mode – we have amended the text in various places to make clear that the experiment was performed in positive mode.

Line 357: Have you a biochemical explanation for this down-regulation?

We don’t have a biochemical explanation, and can only point to previous work which shows that the cytokine storm that is characteristic of severe covid infection is highly disruptive to metabolism. We think that more work is needed before the research community can establish whether salivary metabolite concentrations can be indicative of specific biological processes. 

Lines 364-365: I fully agree. The best choice would be to post-pone the use of predictive models to future studies characterized by a wider cohort of subjects and a selected panel of chemicals fully identified and quantified in targeted mode.

We agree but we do think there is a role for preliminary, and untargeted discovery studies to identify a set of metabolites that are suitable for research in targeted mode.

Reviewer 2 comments

We thank Reviewer 2 for the generally favourable response and hope that the improvements and changes made will be acceptable to Reviewer 2 also.

---

## [Editor Report · Decision Letter 1]

8 Sep 2022

Untargeted saliva metabolomics by liquid chromatography - mass spectrometry reveals markers of COVID-19 severity

PONE-D-22-09481R1

Dear Dr. Matt Spick,

We’re pleased to inform you that your manuscript has been judged scientifically suitable for publication and will be formally accepted for publication once it meets all outstanding technical requirements.

Kind regards,

Tommaso Lomonaco, Ph.D

Academic Editor

PLOS ONE
---

## [Editor Report · Acceptance letter]

13 Sep 2022

PONE-D-22-09481R1 

Untargeted saliva metabolomics by liquid chromatography - mass spectrometry reveals markers of COVID-19 severity 

Dear Dr. Spick:

I'm pleased to inform you that your manuscript has been deemed suitable for publication in PLOS ONE. Congratulations! Your manuscript is now with our production department. 

Kind regards, 

on behalf of

Dr. Tommaso Lomonaco 

Academic Editor

PLOS ONE